# Formulation Development of Mucoadhesive Microparticle-Laden Gels for Oral Mucositis: An In Vitro and In Vivo Study

**DOI:** 10.3390/pharmaceutics12070603

**Published:** 2020-06-29

**Authors:** Hiroomi Sakurai, Yuri Ikeuchi-Takahashi, Ayaka Kobayashi, Nobuyoshi Yoshimura, Chizuko Ishihara, Tohru Aomori, Hiraku Onishi

**Affiliations:** 1Department of Pharmacy, Keio University Hospital, 35 Shinanomachi, Shinjuku, Tokyo 160-8582, Japan; aomori-th@pha.keio.ac.jp; 2Department of Drug Delivery Research, Hoshi University, 2-4-41 Ebara, Shinagawa, Tokyo 142-8501, Japan; y-ikeuchi@hoshi.ac.jp (Y.I.-T.); onishi@hoshi.ac.jp (H.O.); 3Osaka R&D Center, Mitsubishi Chemical Corporation, 2-13-1 Muroyama, Ibaraki, Osaka 567-0052, Japan; kobayashi.ayaka.mb@m-chemical.co.jp (A.K.); yoshimura.nobuyoshi.mg@m-chemical.co.jp (N.Y.); ishihara.chizuko.mw@m-chemical.co.jp (C.I.); 4Hospital Pharmacy Science, Keio University Faculty of Pharmacy, 1-5-30 Shibakoen, Minato, Tokyo 105-8512, Japan

**Keywords:** mucoadhesive microparticles, polyvinyl alcohol, indomethacin, oral mucositis, crystal polymorph

## Abstract

In order to relieve pain due to oral mucositis, we attempted to develop mucoadhesive microparticles containing indomethacin (IM) and gel preparations with IM microparticles that can be applied to the oral cavity. The mucoadhesive microparticles were prepared with a simple composition consisting of IM and polyvinyl alcohol (PVA). Two kinds of PVA with different block properties were used, and microparticles were prepared by heating-filtration and mixing-drying. From the X-ray powder diffraction patterns, differential scanning calorimetry thermograms, and morphological features of the IM microparticles, IM should exist as polymorphic forms in the microparticles. Rapid drug release properties were observed in the IM microparticles. Increased drug retention was observed in IM microparticles containing PVA, and the IM-NK(50) gel, using a common block character PVA and heating-filtration, showed good long-term drug retention properties. In vivo experiments showing significantly higher drug concentrations in the oral mucosa were observed with IM microparticles prepared by heating-filtration, and the IM-NK(50) gel maintained significantly higher drug concentrations in the oral mucosa. From these results, the IM-NK(50) gel may be useful as a preparation for relieving oral mucositis pain.

## 1. Introduction

Oral mucositis is one of the most common complications during chemotherapy and some local radiotherapies that can give significant effect on continuation of the therapy. Oral mucositis results in erythema and ulceration of the non-keratinized mucosa, causing suffering pain, decrease in nutritional intake, and bloodstream infections [1,2,3]. In general, the etiology of chemotherapy-induced mucositis can be divided into two main causes. One primary cause is mucosal inflammation due to the destruction of the cells caused by reactive oxygen species (superoxide or hydroxy radicals) generated by anticancer agents [4,5]. Cells in the oral mucosa also will be exposed to metabolic damage as a result of their uptake of anticancer agents, ends up in blocking healthy oral mucosal cell turnover. The secondary cause is the adhesion of high levels of oral bacteria to the ulcerated surface, causing local infection on the mucosal surface. In combination with metabolic damage and susceptibility to infection caused by anticancer agents, mucositis can become intractable [6]. As yet, no studies of cancer related mucositis with a high evidence level have been published, thus an effective, reliable method of treatment has yet to be established [7,8].

Oral mucositis during chemotherapy or local radiotherapy is characterized by widespread inflammation, which is associated with a high level of pain, and opioid analgesics and nonsteroidal anti-inflammatory drugs (NSAIDs) are introduced [9]. Opioids carry a risk of development of dependence and the side effects of opioids require overall management. The systemic administration of NSAIDs is also limited because of the risk of kidney dysfunction [10]. Topical administration to the oral mucous membrane is an alternative route that specifically aims the affected area and avoid systemic side effects [11,12]. For oral mucositis pain control, oral sprays with indomethacin (IM), an NSAID, were developed as an in-hospital preparation to reduce oral mucositis pain and reduce opioid use [13,14,15]. However, there are limitations in sustained drug delivery by spraying onto the oral mucosa and the stability of the preparation is not sufficient.

Therefore, as a sustained-acting formulation that replaces oral sprays, we attempted to develop mucoadhesive microparticle-laden gels that are applied to the oral cavity to relieve pain caused by oral mucositis after chemotherapy or local radiotherapy. IM was used as an active drug because it has a history of being used in oral sprays for relieving pain in hospital. As mucoadhesive polymers, water-soluble polymers, such as chitosan and carbopol, and Eudragit^®^ as a coating agent, were studied for the development of mucoadhesive buccal films or tablets [16,17,18,19]. Polyvinyl alcohol (PVA) has been used as the carrier polymer in drug delivery devices, the surfactant for forming polymeric drug carriers, and the mucoadhesive carrier [20,21]. Furthermore, it was reported that PVA can aid particle mobility in mucus despite having a mucoadhesive effect [22]. Hence, PVA was used as a mucoadhesive polymer in this study, and mucoadhesive microparticles were prepared using two different preparation methods. The preparation properties were investigated, such as the drug content ratio and drug recovery in the microparticles, the crystal form of IM, drug retention on mucin disks, and drug release properties. Furthermore, with the aim of evaluating drug permeation into buccal tissue, a gel preparation with the mucoadhesive microparticles was developed. The gel preparation properties were investigated, such as drug retention on mucin disks and drug release properties. Drug absorption and drug permeation into the buccal tissue after buccal administration of the gel reparations were evaluated in vivo using mice.

## 2. Materials and Methods

### 2.1. Materials

IM was purchased from FUJIFILM Wako Pure Chemical Corporation (Osaka, Japan). Two grades of PVA, GOHSENOL^TM^ NK-05R and GOHSENOL^TM^ KP-08R, were supplied by Mitsubishi Chemical Corporation (Tokyo, Japan). The characteristics of the PVAs, as stated by the supplier, are shown in Table 1. PVA has both a hydroxyl group and an acetate group in its side chain, and the degree of substitution of the acetate group with the hydroxyl group (degree of saponification) and its distribution state (block character) can be adjusted. In the present study, two PVAs with similar degrees of saponification but different block characters were used to prepare these mucoadhesive microparticles. Mucin from porcine stomachs was obtained from Sigma-Aldrich Co. (St. Louis, MO, USA). All other chemicals were procured commercially at the highest purity grade available.

### 2.2. Preparation of IM Microparticles

The composition of the mucoadhesive microparticles of IM is shown in Table 2. The mucoadhesive microparticles were prepared using two different preparation methods: the heating-filtration method and mixing-drying method. IM microparticles (IM-NK(100), IM-NK(50), IM-KP(100), IM-KP(50)) were prepared using the heating-filtration method as follows: 100 mg IM was dissolved in 2 mL ethanol, and 5 mL of 2% or 1% PVA aqueous solution was gradually added to the IM/ethanol solution. The mixture suspension was agitated, and the ethanol was subsequently evaporated using N_2_ gas reflux. It is known that PVA (NK-05R and KP-08R) has a cloud point. A photograph of the 1% PVA solutions heated to 70 °C is shown in Figure 1. When the PVA (NK-05R and KP-08R) solutions were heated to 70 °C, the solution became cloudy because the PVA precipitated. Thus, after the ethanol had evaporated from the IM/PVA suspension, the suspension was heated to 70 °C and agitated for 10 min with the aim of precipitating PVA. The suspension was filtered with a membrane filter (Durapore^®^, Merck Millipore Ltd., Burlington, MA, USA; pore size = 0.45 μm). The residue on the membrane filter was dried at reduced pressure for 24 h to obtain an IM/PVA solid mixture. The IM/PVA solid mixture was ground by stirring with 5–7 dry ice pellets (5–10 mm diameter), then sieved (mesh size: 840 μm) to obtain the microparticles. In addition, IM microparticles (IM-NK(Dry), IM-KP(Dry)) were prepared using the mixing-drying method as follows: 100 mg IM was dissolved in 2 mL ethanol, and 5 mL of aqueous solution containing 12 mg dissolved PVA was added gradually to the IM/ethanol solution. The mixture suspension was agitated, and the ethanol was evaporated from the mixture using N_2_ gas reflux. After the ethanol had evaporated, the mixture suspension was poured into a balance dish. The balance dish was dried at reduced pressure for 48 h to obtain an IM/PVA solid mixture. Because the IM/PVA solid mixture was able to be sieved without grinding, the mixture was sieved (mesh size: 840 μm) to obtain the microparticles. In order to evaluate the effect of adding the PVA to the microparticles, IM microparticles without PVA were also prepared. IM microparticles without PVA were prepared as follows: 100 mg IM was dissolved in 2 mL ethanol, and 5 mL of purified water was gradually added to the IM/ethanol solution. The IM suspension was agitated, and the ethanol was subsequently evaporated from the mixture by N_2_ gas reflux. After the ethanol had evaporated, the suspension was filtered with a membrane filter (Durapore^®^, Merck Millipore Ltd.; pore size = 0.45 μm). The residue on the membrane filter was dried under reduced pressure for 24 h to obtain IM solid. The IM solid was able to be sieved (mesh size: 840 μm) without grinding, and the IM microparticles without PVA were obtained.

### 2.3. Drug Content Ratio, Drug Recovery, PVA Content Ratio, and PVA Recovery

The drug content ratio (D_cont_) was calculated as follows:
D_cont_ (%) = (W_d_/W_p_) × 100.(1)


The drug recovery (D_reco_) was calculated as follows:
D_reco_ (%) = (W_d_/W_id_) × 100,(2)
W_d_ is the amount of IM in the preparation, W_p_ is the amount of the preparation, and W_id_ is the amount of initial IM.

The PVA content ratio (P_cont_) was calculated as follows:
P_cont_ (%) = {(W_p_ − W_d_)/W_p_} × 100.(3)


The PVA recovery (P_reco_) was calculated as follows:
P_reco_ (%) = {(W_p_ − W_d_)/W_iPVA_} × 100,(4)
where W_iPVA_ is the amount of initial PVA.

### 2.4. Viscosity Measurements of PVA Solutions

The viscosities of the 1% and 2% PVA solutions were measured at 20 °C, 30 °C, 40 °C, 50 °C, and 60 °C using a cone and plate viscometer (TVE-35, Toki Sangyo Co., Ltd. (Tokyo, Japan)) equipped with a 1°34′ × R24 cone rotor. The shear stress (Pa) was measured by increasing and decreasing the shear rate from 0.766 to 383 (s^−1^), and the viscosity (mPa·s) was calculated from the rheogram. In this study, the viscosity of the shear rate at 383 (s^−1^), where it was most stable, was used for comparison of the solution viscosities.

### 2.5. Morphological Features of the Microparticles 

The morphologies of the IM bulk powder and the IM microparticles without PVA were characterized using an optical microscope (OLYMPUS BX51, Olympus Corporation, Tokyo, Japan) equipped with a microscope digital camera (Visualix V500FL, Visualix K.K., Hyogo, Japan). A 40× magnification objective lens (UplanApo, Olympus Corporation) was used. The morphologies of the IM microparticles (IM-NK(50), IM-KP(50), IM-NK(Dry), IM-KP(Dry)) were characterized using a scanning electron microscope (SEM). The JSM-6060LA scanning electron microscope (JEOL Ltd., Tokyo, Japan) was used to obtain SEM images.

### 2.6. Particle Size Measurements of Microparticles

IM microparticles were observed using an optical microscope (OLYMPUS BX51, Olympus Corporation) equipped with a microscope digital camera (Visualix V500FL, Visualix K.K.). The Feret diameter of 100 particles of the IM microparticles was measured randomly by microscopic observations [23].

### 2.7. X-ray Powder Diffraction

X-ray powder diffraction (XRPD) patterns of the IM microparticles were determined using a 9 kW SmartLab diffractometer (Rigaku Corporation, Tokyo, Japan) with a rotating anode at room temperature. The voltage and amperage were set at 45 kV and 200 mA, respectively. Each sample was scanned between 5° and 40° in 2 θ with a step size of 0.02 and scan speed of 20°/min.

### 2.8. Differential Scanning Calorimetry

The thermal properties of the IM microparticles were determined using differential scanning calorimetry (DSC) (Thermo plus EVO II DSC8230, Rigaku Corporation). Samples weighing approximately 2 mg were heated in a sealed aluminum pan at a constant heating rate of 10 °C/min from 25 °C to 180 °C with a nitrogen purge (100 mL/min) [24].

### 2.9. Drug Retention Test

Mucin disks were prepared using the method of Tsuchiya et al. [25]. Four-hundred microliters of 10% mucin solution was spread on a 25-mm diameter filter paper. The filter paper was dried at room temperature for 24 h. An artificial saliva solution (pH 6.8) consisting of 0.8% NaCl, 0.019% KH_2_PO_4_, and 0.238% Na_2_HPO_4_ was prepared. The mucin disk was moistened with artificial saliva, and 3 mg of IM bulk powder or the IM microparticles were mounted on the mucin disk, and the mucin disk was dried at room temperature for 5 min. Each mucin disk with sample was fixed to a slide glass with a clip and immersed in 150 mL of artificial saliva at 37 °C and incubated with shaking at 50 rpm. Mucin disks were taken out at 5, 10, and 15 min and the amount of IM remaining on the disk was measured using high-performance liquid chromatography (HPLC). The HPLC system consisted of an LC-6AD pump (Shimadzu Corporation, Kyoto, Japan) and a Chromato-PRO (Run Time Corporation, Tokyo, Japan) equipped with a Capcell Pak C18 MG II column (4.6 × 250 mm, OSAKA SODA CO., LTD., Osaka, Japan) and an SPD-20AV UV detector (Shimadzu Corporation). Chromatography was carried out at 40 °C. The mobile phase comprised 60% (*v*/*v*) acetonitrile in 0.02 M sodium acetate buffer adjusted to pH 3.6 using orthophosphoric acid. The flow rate was 1 mL/min. The detection wavelength was 320 nm.

### 2.10. Drug Release Properties

The in vitro drug release from the IM bulk powder and IM microparticles was examined using a Franz diffusion cell. Drug release experiments were performed with a water: 50 mM phosphate buffer (KH_2_PO_4_-NaOH, pH 7.2) (4:1) as the dissolution media in accordance with a method for dissolution tests for IM capsules in the Pharmacopoeia of Japan (JP) 17. Membrane filter (Durapore^®^, Merck Millipore Ltd.; pore size = 0.45 μm) were mounted between the donor and receiver compartments of the Franz diffusion cell (exposed area = 1.77 cm^2^). The receiver compartment was filled with dissolution media. The dissolution media in the receiver chamber was stirred with a magnetic stirrer at a speed of 700 rpm and maintained at 37 °C. Three milligrams of the sample were placed in the donor compartment, and at predetermined time intervals, a 0.3-mL sample of the dissolution media was taken and replaced with fresh medium. The sample solution was diluted with fresh medium and absorbance at 320 nm was measured using a UV spectrophotometer (UV-1800, Shimadzu Corporation) in order to evaluate the quantity of the drug released.

### 2.11. Formulation of Gel Preparations with IM Microparticles

To evaluate drug permeation into the buccal tissue, a gel preparation with mucoadhesive microparticles were prepared. A gel was prepared by gelling an aqueous gelatin solution with added glycerin, i.e., purified water and glycerin (1:1) were mixed and 5% gelatin was added to the mixture solution. The solution was heated to 80 °C and stirred until the gelatin dissolved. Eight-hundred microliters of the mixture solution was poured into a balance dish (22 mm × 22 mm), and the balance dish was allowed to stand at 2 °C to prepare a gel. A gel preparation was prepared by uniformly dispersing microparticles on the prepared gel, which was about 1 mm thick.

### 2.12. Drug Retention Test Using Gel Preparations

Mucin disks were prepared in the same manner as the retention test for microparticles. A gel preparation was prepared by uniformly dispersing 1 mg of IM bulk powder or IM microparticles on the prepared gel (11 mm × 11 mm). The mucin disk was moistened with artificial saliva, and the gel preparation was set on the mucin disk so that the IM microparticles were in contact with the mucin disk. The mucin disk with the gel preparation was fixed to a slide glass with a clip and immersed in 150 mL of artificial saliva at 37 °C and incubated with shaking at 50 rpm. Mucin disks were taken out at 5, 10, and 15 min, and the amount of IM remaining on the disk was measured by HPLC. The HPLC conditions were the same as the retention test for microparticles.

### 2.13. Drug Release Properties from Gel Preparations

A gel preparation was prepared by uniformly dispersing IM bulk powder or IM microparticles (0.75 mg IM) on the prepared gel (11 mm × 11 mm). The gel preparation was set in the donor compartment so that the IM microparticles were in contact with the membrane filter. The drug release experiment was performed in the same manner as the drug release test with microparticles.

### 2.14. In Vivo Retention

The animal protocols in this study were approved by the issuing committee (Committee on the Care and Use of Laboratory Animals of Hoshi University), which is accredited by the Ministry of Education, Culture, Sports, Science, and Technology, Japan, as conforming with the Guide for the Care and Use of Laboratory Animals (Approval No. 30-134, dated 17 December 2018). 

Forty-eight male Jcl:ICR mice were induced and all were between 6 and 7 weeks of age. Mice were housed in temperature-controlled cages with a 12-h light–dark cycle and were given free access to water and a normal chow diet until the experiments were conducted. The mice were divided into 12 groups with 4 mice in each group. Six groups were used for 1 h application experiments. The remaining 6 groups were used for 3 h application experiments. IM bulk powder, IM microparticles without PVA, IM-NK(50), IM-KP(50), IM-NK(Dry), or IM-KP(Dry) (1.5 mg IM) was uniformly dispersed on the prepared gel (11 mm × 11 mm) and the gel was divided into six pieces. One piece of the gel preparation (250 μg IM/body) was administrated to the oral cavity of a mouse under temporary isoflurane anesthesia. Water was freely accessible, but food was fasted during the test. After 1 h and 3 h, blood was collected from the postcava under isoflurane anesthesia, and the buccal mucosa was removed after mice were euthanized using excess anesthesia.

Plasma samples were isolated from the blood by centrifugation at 1000× *g* for 15 min. The plasma concentrations of the IM were determined using HPLC. Mefenamic acid was used as an internal standard. Each 100 μL plasma sample was dispensed into a centrifuge tube. Next, 160 μL of methanol and mefenamic acid in methanol (10 μg/mL, 40 μL) were added. The samples were agitated for 4 min and then centrifuged at 1000× *g* for 15 min. The supernatant was transferred to a clean centrifuge tube and concentrated under a nitrogen flow. A further 100 μL of methanol was added to the residue, and the mixture was agitated for 4 min and then centrifuged at 1000× *g* for 5 min. The supernatant was filtered, and 20 μL of the filtrate was used for HPLC analysis. 

Excised oral mucosa was weighed and the surface was washed with physiological saline solution. The tissue sample was homogenized by 2 mL of a 75% (*v*/*v*) methanol solution, and the extract solution was centrifuged at 1000× *g* for 10 min, and the supernatant was filtered through a membrane filter (0.45 μm). The filtrate solution (20 μL) was used for HPLC analysis.

The HPLC conditions were the same as for the retention test for microparticles.

### 2.15. Statistical Analysis

A one-way analysis of variance with Dunnett’s multiple comparison test was performed to compare the drug retention properties and the tissue and the plasma drug concentrations on the IM microparticles against IM bulk powder. Data were considered significantly different when the *p*-value was less than 0.05.

## 3. Results and Discussion

### 3.1. Characterization of Microparticles

The D_cont_, D_reco_, P_cont_, P_reco_, and Feret diameter of the IM microparticles are shown in Table 3. D_reco_ showed high values in all IM microparticles. In IM microparticles prepared using the heating-filtration method, IM-NK(50) and IM-KP(50) showed higher P_cont_ values compared with IM-NK(100) and IM-KP(100). In Figure 2, the viscosities of the 1% and 2% PVA solutions are plotted against the temperature at the time of measurement. The 2% PVA solutions showed higher viscosity than the 1% PVA solutions in the temperature range measured. During preparation using the heating-filtration method, the IM/PVA suspension was heated to 70 °C, then the suspension was filtered using a membrane filter at room temperature. If the filtration time is prolonged, the precipitated PVA will be filtered out, and P_cont_ will decrease. Because the filtration time was shorter using the lower viscosity 1% PVA solution, the P_cont_ of IM-NK(50) and IM-KP(50) should be increased. Traditionally, the effect of temperature on the Newtonian viscosity or its apparent viscosity at a specified shear rate of a liquid food has been described using the Arrhenius equation [26]. The temperature–viscosity model can be written in the following form:
Ln[*μ*(*T*)/*μ*(*T*_ref_)] = *E*_a_/*R* (1/*T* − 1/*T*_ref_),(5)
where *μ*(*T*) is the Newtonian or apparent viscosity at an absolute temperature *T* (in K) and *μ*(*T*_ref_) at an arbitrary absolute reference temperature *T*_ref_. *E*_a_ is the energy of activation and *R* is the universal gas constant. A linear Ln[μ(*T*)] vs. 1/*T* plot is one of the implications of Equation (5). Figure 3 shows graphs of the natural logarithm of the viscosity in Figure 2 plotted against the reciprocal of the absolute temperature (1/*T*). In the NK-05R solution, the increase in the slope was shown from 313 K (40 °C) to 323 K (50 °C). In the KP-08R solution, an increase in the slope was shown from 303 K (30 °C) to 313 K (40 °C). Therefore, it was considered that the precipitation of NK-05R mainly occurred from 313 K (40 °C) to 323 K (50 °C), and the precipitation of KP-08R mainly occurred from 303 K (30 °C) to 313 K (40 °C). KP-08R had a lower cloud point than NK-05R, suggesting that it precipitated at lower temperatures. From these results, because IM-NK(50) and IM-KP(50) showed higher P_cont_ values than IM-NK(100) and IM-KP(100), IM-NK(50) and IM-KP(50) were used for the experiments thereafter.

Because the P_cont_ of IM-NK(50) and IM-KP(50) was about 10%, the amount of PVA added to prepare IM-NK(Dry) and IM-KP(Dry) was set to 12 mg. From Table 3, the P_cont_ of IM-NK(Dry) and IM-KP(Dry) was about 10%. Regarding the Feret diameter, IM microparticles using KP-08R tended to be larger than IM microparticles using NK-05R. Because KP-08R has enhanced blocking properties compared with NK-05R, it was considered that the intermolecular interactions of KP-08R increased, and the particle size tended to be larger.

Optical micrographs of IM bulk powder and IM microparticles without PVA, as well as SEM images of IM-NK(50), IM-KP(50), IM-NK(Dry), and IM-KP(Dry), are shown in Figure 4. From the optical micrographs (Figure 4a,b), the morphological features of IM microparticles without PVA were different from the IM bulk powder. The XRPD patterns of IM microparticles are shown in Figure 5. The XRPD spectrum of IM microparticles without PVA was different from the IM bulk powder. DSC thermograms of IM microparticles are shown in Figure 6. From the DSC thermograms, the melting points of IM bulk powder and IM microparticles without PVA were 161 °C and 155 °C, respectively. Previous reports suggested that the IM crystal form, showing a peak at 161 °C, was the γ form, and that the IM crystal form showing a peak at 155 °C was the α form [27,28]. The DSC results were also consistent with the results of the XRPD spectrum and the morphological features [29]. From the SEM images of IM-NK(50), IM-KP(50), IM-NK(Dry), and IM-KP(Dry) (Figure 4c–f), it was assumed that the IM crystals as IM microparticles without PVA were coated and bound by PVA. Furthermore, there were more pores in IM-NK(Dry) and IM-KP(Dry) than in IM-NK(50) and IM-KP(50). The pores in IM-NK(50) and IM-KP(50) should decrease during the filtration process. The XRPD spectra of IM-NK(50), IM-KP(50), IM-NK(Dry), and IM-KP(Dry) were similar to that of IM microparticles without PVA. From the DSC thermograms, the melting points of IM-NK(50), IM-KP(50), IM-NK(Dry), and IM-KP(Dry) were 154 °C, 154 °C, 153 °C, and 153 °C, respectively. From the results of the XRPD spectra and the melting points, the existence of the α form of IM in IM microparticles was suggested. However, the decrease in the intensity of the XRPD spectra suggested that the crystallinity of IM in IM microparticles containing PVA may have decreased.

### 3.2. Drug Retention and Drug Release Properties of IM Microparticles

The drug retention ratio after the application of IM microparticles to mucin disks is shown in Figure 7. The drug retention ratios of IM microparticles containing PVA were increased compared with IM bulk powder. Hence, it was suggested that PVA covering the IM microparticles can interact with the mucin disk [30,31]. However, the type of PVA and the preparation method of IM microparticles did not significantly affect the drug retention ratio. During the examinations, it was possible that the resistance of the water was strong and a difference between the IM microparticles containing PVA was hardly observed. The drug release profiles from the IM microparticles are shown in Figure 8. More rapid drug release from IM microparticles without PVA was observed compared with IM bulk powder. The fact that IM exists as a polymorph in IM microparticles without PVA should be considered. Rapid drug release from IM microparticles containing PVA was also observed compared with IM bulk powder. This was also considered to be due to polymorphism and a decrease in the crystallinity of IM. Although differences in the drug release profiles due to the type of PVA were not observed, the more rapid drug release from IM-NK(Dry) and IM-KP(Dry) than IM microparticles without PVA was observed. This was considered to be due to the wettability of IM microparticles by water being improved with PVA [32,33]. On the other hand, drug release from IM-NK(50) and IM-KP(50) was slower than from IM microparticles without PVA. It was considered that IM-NK(50) and IM-KP(50) formed a packed mass with fewer pores through intermolecular interaction with PVA.

### 3.3. Drug Retention and Drug Release Properties of Gel Preparations

IM microparticles containing PVA showed retention properties to the mucin disk. Therefore, drug release was improved compared with the IM bulk powder, and thus these have potential to be useful as mucoadhesive microparticles. However, because poor usability of the IM microparticles for administration to the oral cavity was suggested, gel preparations with IM microparticles were prepared. The drug retention ratio after application of the gel preparations to mucin disks is shown in Figure 9. At 5 min, IM-KP(50) gel and IM-KP(Dry) gel showed significantly higher values than IM bulk powder gel. However, at 10 min, only IM-NK(50) gel showed a significantly higher value than IM bulk powder. At 15 min, no significant difference was observed, but higher values were shown for IM bulk powder gel and IM microparticles without the PVA gel compared with IM microparticles containing PVA gel. It was suggested that the use of PVA (KP-08R) increased the initial affinity between the microparticles and mucin. In IM bulk powder gel and IM microparticles without PVA gel, it was considered that a proportion of the drug was dissolved in the gel base, and the dissolved drug was retained on the mucin disk. Drug release profiles from the gel preparations are shown in Figure 10. In IM bulk powder gel and IM microparticles without PVA gel, drug release after 1 h was suppressed. It was considered that some of the drug in IM bulk powder and IM microparticles without PVA was dissolved in the gel base, and the drug dissolved in the gel base was hardly released. On the other hand, drug release from IM microparticles contained in the PVA gel was over 90% at 4 h. Hence, it was considered that the drug may be prevented from dissolving in the gel base by PVA coating. Drug release from IM-NK(Dry) gel and IM-KP(Dry) gel was slightly faster than that from IM-NK(50) gel and IM-KP(50) gel. This result was consistent with the results of drug release from the IM microparticles, and was considered to be due to the fact that many pores were observed in IM-NK(Dry) and IM-KP(Dry).

### 3.4. In Vivo Experiments

The drug concentrations in the oral mucosa and in plasma after application of the gel preparations to the oral cavity of mice are shown in Figure 11. For the tissue concentration at 1 h after administration to the oral mucosa, IM-NK(50) gel and IM-KP(50) gel showed significantly higher values than IM bulk powder gel, and at 3 h, IM-NK(50) gel showed a significantly higher value than IM bulk powder gel. For the in vitro drug retention tests, although the IM-KP(50) gel showed a significantly higher value at 5 min, the higher value did not continue until 10 min. Because PVA (KP-08R) was suggested to precipitate at 30–40 °C due to the cloud point, KP-08R may be precipitated at body temperature and its mucoadhesive properties reduced gradually. From these results, KP-08R had high affinity with the oral mucosa, but the affinity may be not maintained at body temperature. On the other hand, because NK-05R has high affinity with mucous membranes and the cloud point was higher than KP-08R, the affinity with the mucosa should be maintained and the highest tissue concentration was shown at 3 h. IM-NK(Dry) gel and IM-KP(Dry) gel did not show significantly higher tissue concentrations than IM bulk powder gel. IM-NK(Dry) and IM-KP(Dry) had more pores from the SEM images and released the drug more rapidly compared with IM-NK(50) and IM-KP(50).

Hence, in the case of IM-NK(Dry) gel and IM-KP(Dry) gel, it was considered that the drug diffused into the oral cavity, thereby suppressing permeation of the drug into the oral mucosa. For IM microparticles without PVA gel as well, it was considered that the mucosal permeability may have decreased due to drug diffusion into the oral cavity. IM-NK(50) and IM-KP(50) prepared by the heating-filtration method formed packed microparticles with fewer pores, and the IM microcrystals in IM-NK(50) and IM-KP(50) may be tightly covered by PVA. Such microparticle structures may have effectively acted to promote drug permeability through the mucosa. For drug concentrations in plasma, none of the IM microparticle gels showed significantly higher values compared with IM bulk powder gel. The relatively lower plasma drug concentrations were confirmed in IM microparticles without PVA gel compared with the other gels. This was probably due to the fact that the mouse felt the bitterness of the drug and the discomfort of the formulation due to the difference in morphology of the microparticles, and then spat out the preparation.

From these results, among the IM microparticles prepared in this study, IM-NK(50) gel showed the highest drug concentration in the oral mucosa, and the drug concentrations in plasma were not significantly different from that of IM bulk powder gel. Hence, it is suggested that IM-NK(50) gel was useful as a preparation for relieving oral mucositis pain.

## 4. Conclusions

Because there are no satisfactory preparations to relieve pain due to oral mucositis associated with chemotherapy and/or local radiotherapy, we attempted to develop mucoadhesive microparticle-laden gels that can be applied to the oral cavity. The mucoadhesive microparticles were prepared with a simple composition consisting of IM and PVA. From the results of the DSC thermograms, XRPD spectra, and morphological features, it was suggested that IM in the IM microparticles existed in a polymorphic form (α form). Increased drug retention ratios of the IM microparticles containing PVA were observed; therefore, it was suggested that PVA covering the IM microparticles may interact with the mucin disk. Rapid drug release from the IM microparticles containing PVA was observed compared with IM bulk powder, and more rapid drug release from IM-NK(Dry) and IM-KP(Dry) was observed than from IM-NK(50) and IM-KP(50). In IM-NK(50) and IM-KP(50), it was considered that a more packed mass with fewer pores was prepared by the intermolecular interaction with PVA. During the drug release test of the gel preparations, some drug in IM bulk powder and IM microparticles without PVA may be dissolved in the gel base. In IM microparticles containing PVA, the drug in the IM microparticles containing PVA may be prevented from dissolving in the gel base by the PVA coating. In in vivo experiments, sustained higher drug concentrations in the oral mucosa were observed in the IM-NK(50) gel. As for a reason why the high tissue drug concentrations did not last in the IM-KP(50) gel, it was considered that KP-08R did not maintain an affinity with the oral mucosa at body temperature due to the precipitation cloud point. The structures of IM-NK(50), where the IM microcrystals were covered with PVA and packed, may have effectively acted to promote drug permeability through the mucosa. Because the drug concentrations in the plasma after the administration of IM-NK(50) gel were equivalent to that after the administration of the IM bulk powder gel, it was suggested that IM-NK(50) gel is acceptable for relieving oral mucositis pain.

## Figures and Tables

**Figure 1 pharmaceutics-12-00603-f001:**
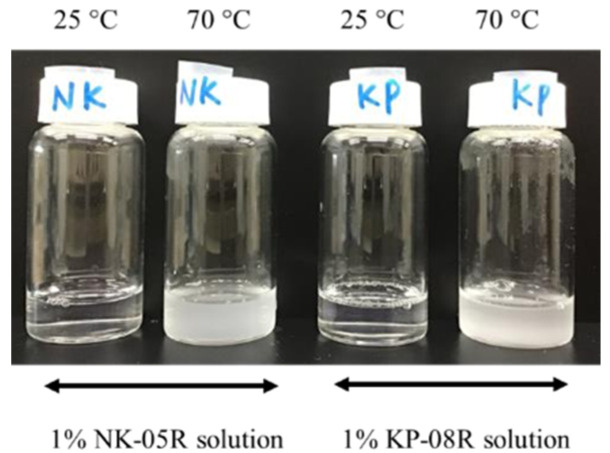
Photograph of the 1% PVA solutions at 25 °C and 70 °C.

**Figure 2 pharmaceutics-12-00603-f002:**
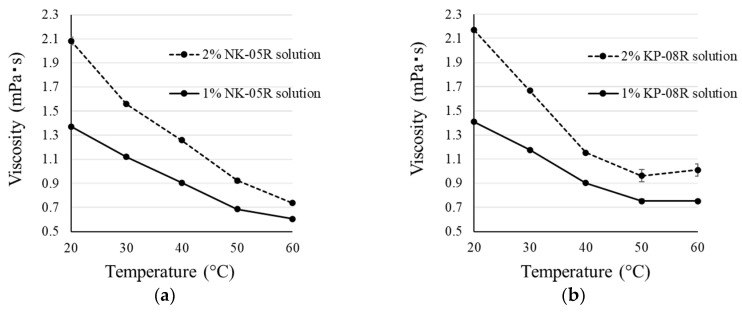
Viscosities of the 1% and 2% PVA solutions plotted against temperature at the measurement: (**a**) NK-05R solution; (**b**) KP-08R solution. Each point represents the mean ± SD (*n* = 3).

**Figure 3 pharmaceutics-12-00603-f003:**
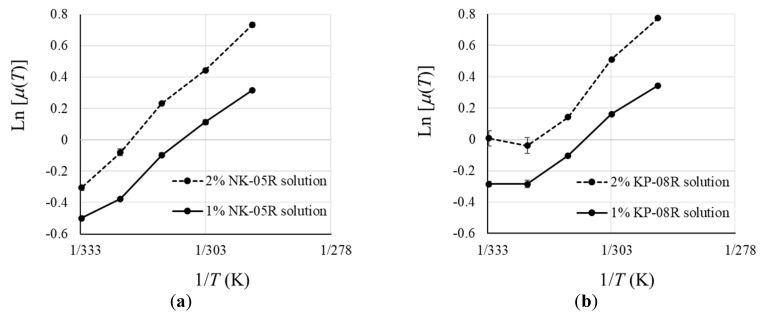
Graphs of the Natural Logarithm of Viscosity Plotted Against the Reciprocal of the Absolute Temperature (1/T): (**a**) NK-05R Solution; (**b**) KP-08R Solution. Each Point Represents the Mean ± SD (*n* = 3).

**Figure 4 pharmaceutics-12-00603-f004:**
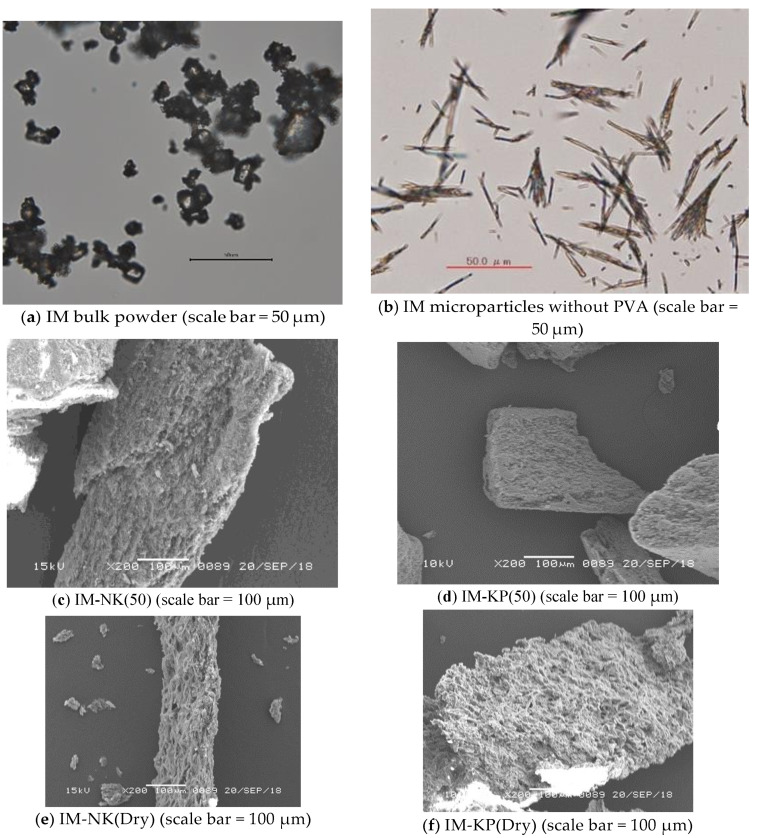
(**a**,**b**) Optical Micrographs of IM Bulk Powder and IM Microparticles Without PVA; (**c**–**f**) Scanning Electron Microscopy Images of IM-NK(50), IM-KP(50), IM-NK(Dry), and IM-KP(Dry).

**Figure 5 pharmaceutics-12-00603-f005:**
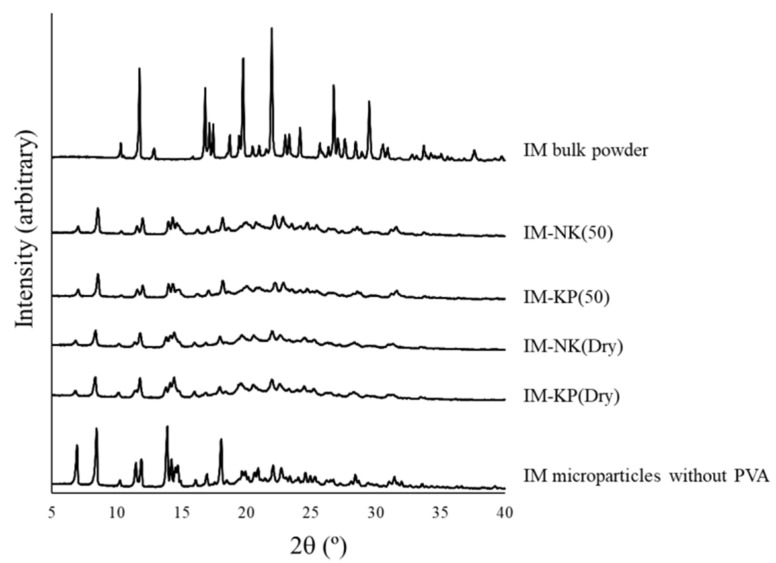
XRPD Patterns of IM Microparticles.

**Figure 6 pharmaceutics-12-00603-f006:**
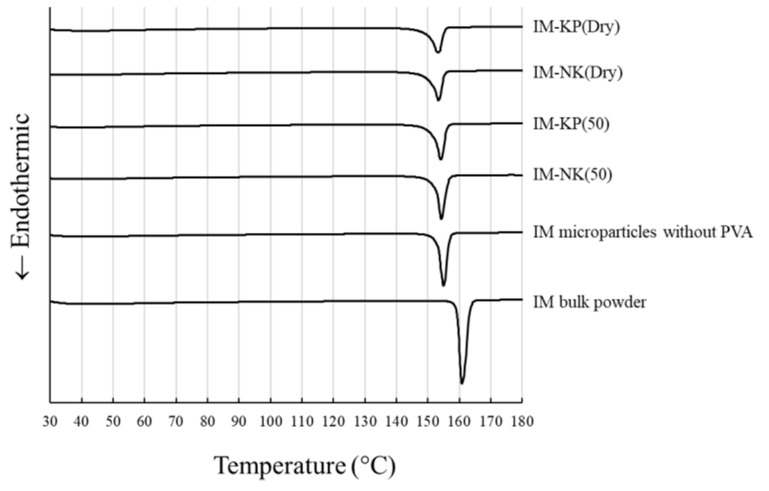
DSC Thermograms of the IM Microparticles.

**Figure 7 pharmaceutics-12-00603-f007:**
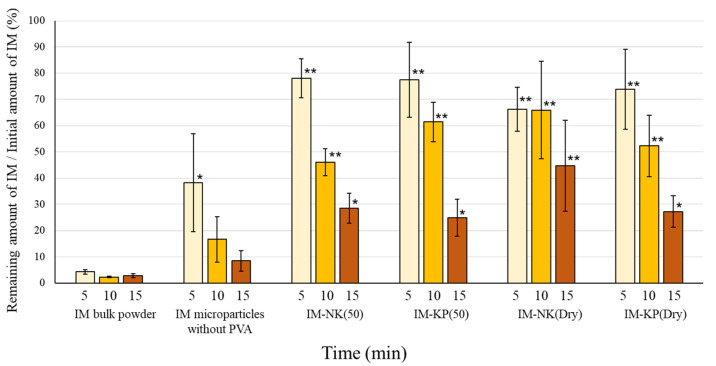
Drug Retention Ratio After the Application of IM Microparticles to Mucin Disks. Each Value Represents the Mean ± SD (*n* = 3). ** *p* < 0.01, * *p* < 0.05; Dunnett’s Test Was Performed. IM Microparticles > IM Bulk Powder.

**Figure 8 pharmaceutics-12-00603-f008:**
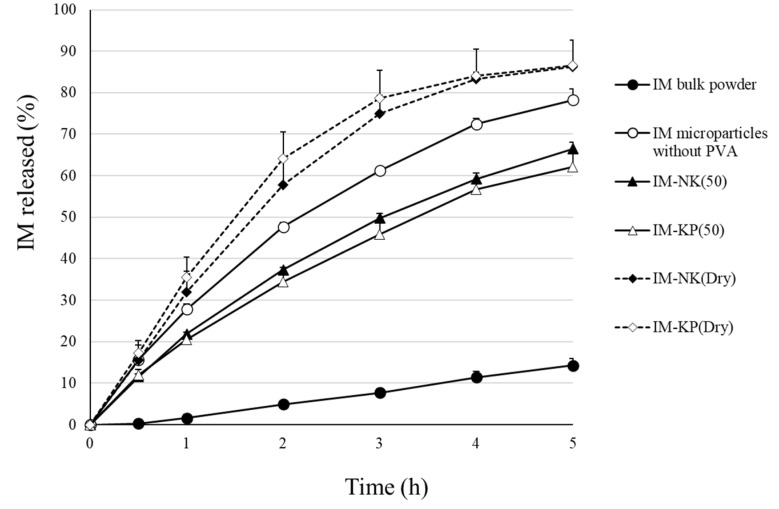
Drug Release Profiles from the IM Microparticles. Each Point Represents the Mean ± SD (*n* = 3).

**Figure 9 pharmaceutics-12-00603-f009:**
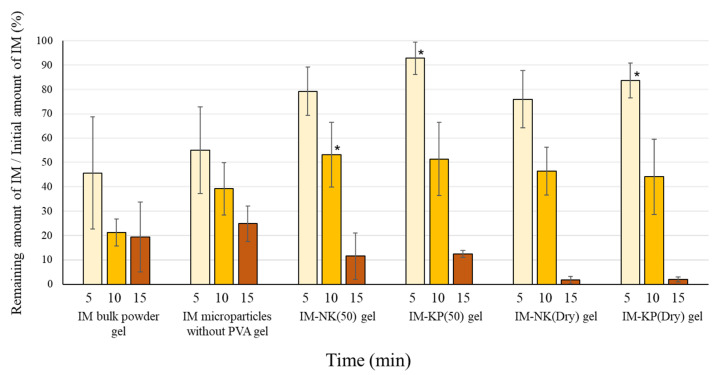
Drug Retention Ratio After Application of the Gel Preparations to the Mucin Disks. Each Value Represents the Mean ± SD (*n* = 3). * *p* < 0.05; Dunnett’s Test Was Performed. IM Microparticle Gels > IM Bulk Powder Gel.

**Figure 10 pharmaceutics-12-00603-f010:**
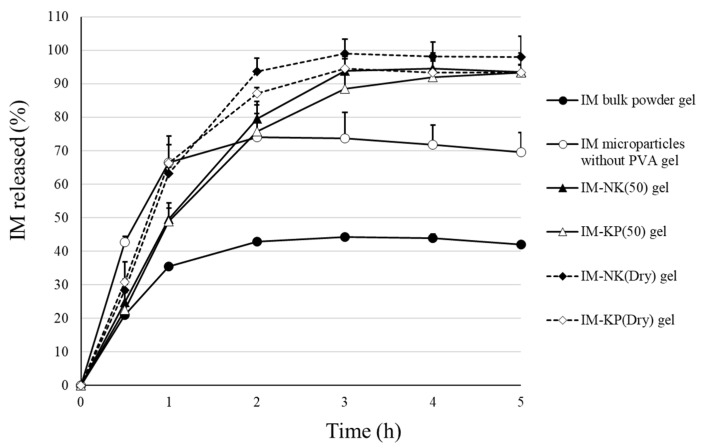
Drug Release Profiles from the Gel Preparations. Each Point Represents the Mean ± SD (*n* = 3).

**Figure 11 pharmaceutics-12-00603-f011:**
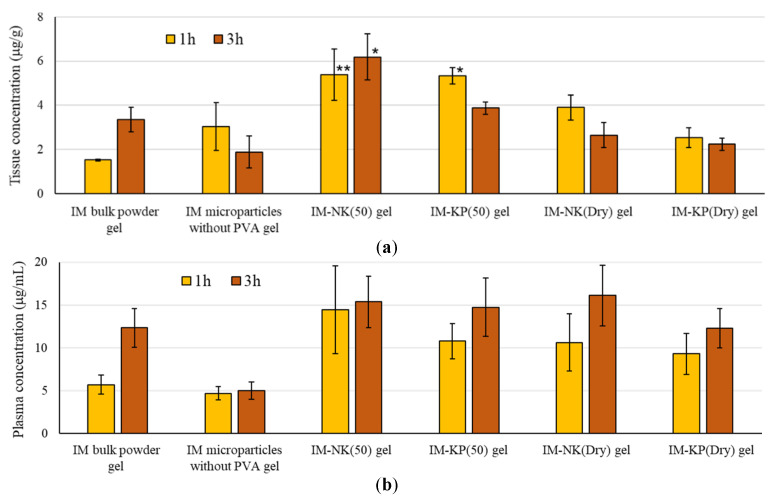
(**a**) Drug Concentrations in the Oral Mucosa and (**b**) Drug Concentrations in the Plasma After Application of Gel Preparations to the Oral Cavity of Mice. Each Value Represents the Mean ± SE (*n* = 4). ** *p* < 0.01, **p* < 0.05; Dunnett’s Test Was Performed. IM Microparticle Gels > IM Bulk Powder Gel.

**Table 1 pharmaceutics-12-00603-t001:** Characteristics of the PVAs.

PVA	Saponification Degree (mol%)	Estimated Degree of Polymerization	Block Character	Cloud Point (°C)
NK-05R	71.0–75.0	500	Common-product	30 (approximately)
KP-08R	71.0–73.5	700	Block-up-product	30 (approximately)

**Table 2 pharmaceutics-12-00603-t002:** Composition of the Mucoadhesive Microparticles of the Indomethacin (IM) Prepared by the Heating-Filtration Method and Mixing-Drying Method.

Heating-Filtration Method			
**Formulation**	**IM (mg)**	**PVA (NK-05R) (mg)**	**PVA (KP-08R) (mg)**
IM-NK(100)	100	100	-
IM-NK(50)	100	50	-
IM-KP(100)	100	-	100
IM-KP(50)	100	-	50
**Mixing-Drying Method**			
**Formulation**	**IM (mg)**	**PVA (NK-05R) (mg)**	**PVA (KP-08R) (mg)**
IM-NK(Dry)	100	12	-
IM-KP(Dry)	100	-	12

**Table 3 pharmaceutics-12-00603-t003:** Drug Content Ratio, Drug Recovery, PVA Content Ratio, PVA Recovery, and Feret Diameter of IM Microparticles.

Formulation	D_cont_ (%)	D_reco_ (%)	P_cont_ (%)	P_reco_ (%)	Feret Diameter (μm)
IM-NK(100)	90.1 ± 2.6	89.4 ± 3.8	9.9 ± 2.6	9.8 ± 2.5	269.9 ± 122.7
IM-KP(100)	93.9 ± 2.3	91.0 ± 4.9	6.1 ± 2.3	5.8 ± 2.1	312.1 ± 188.0
IM-NK(50)	89.3 ± 2.6	92.2 ± 5.7	10.7 ± 2.6	21.8 ± 4.9	282.5 ± 88.9
IM-KP(50)	89.8 ± 1.2	94.8 ± 3.9	10.2 ± 1.2	22.2 ± 4.4	359.0 ± 135.9
IM-NK(Dry)	90.1 ± 0.5	84.9 ± 2.6	9.9 ± 0.5	77.5 ± 2.3	261.2 ± 65.9
IM-KP(Dry)	90.2 ± 0.5	79.7 ± 7.0	9.8 ± 0.5	71.4 ± 5.5	265.9 ± 59.2

Values except for the Feret diameter are described as the mean ± SD (*n* = 3). Feret diameters are described as the mean ± SD (*n* = 100).

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
