# Peer review of "Formulation Development of Mucoadhesive Microparticle-Laden Gels for Oral Mucositis: An In Vitro and In Vivo Study"

_pharmaceutics, 2020, doi:10.3390/pharmaceutics12070603_

Round 1
Reviewer 1 Report
The manuscript described a gel formulation and mucoadhesive microparticles for reducing oral pain and presented some interesting results. There are points to consider before publication:
- Title: the title included mucoadhesive microparticles twice, please consider to be more concise.
- Introduction: The paragraph is too long. Consider breaking up.
- Line 63-75: These information including Table 1 should be in Methods.
- The introduction needs further information on: other research in using mucoadhesive formulations for oral mucosa delivery, other polymers for making mucoadhesive microparticles, why select PVA?
- Introduction: The aim of the article is needed
- Methods: any references used in preparation of the microparticles?
- Section 2.4, why choose this shear rate?
- Section 2.6, can laser diffraction be used to measure particle size?
- Section 2.12 how much MPs needed on the gel?
- Section 2.13 how much IM powder needed on the gel?
- Section 2.11 move this section together with other formulation sections
- Section 2.12 and 2.13 can be combined with 2.9 and 2.10 respectively.
- Section 2.14 subtitle should reflect the actual work, e.g. in vivo retention etc
- Results and discussion: Need better clarification on the polymorphism and amorphous content. The two polymorphs, which one is present in which formulation? Which formulation has or has higher amorphous content? Do you know the solubility of each polymorph?
- The morphology of the MPs are rather irregular/needle like. It is difficult to call them MPs rather than aggregates. How would this affect drug release and mouth feel?
- The taste of the gel might be an issue. What is the strategy of taste masking?
- Conclusion is too long and included results and discussion. It should be more concise and only summarise the main findings.
- Need extensive review in English language.
Reviewer 2 Report
The current manuscript by Sakurai et al., is based on an interesting concept backed my some valuable results. However, the present form of the manuscript is poorly presented. I can’t recommend the present form for publication. I have listed following suggestion and would like authors to adopt.
Title:
It’s not only lengthy but authors are repeating themselves please consider modification. What about “Formulation development of mucoadhesive microparticle laden gels for oral mucositis: an in-vitro and in-vivo studies”
Abstract:
- It simply lack rationale, like what is your research question, please don’t present too much of your findings but rather focus on rationale and explain briefly how your results had made progress towards addressing this question.
- Proofreading is strongly advised.
Introduction:
- It start-off well however, authors lost their direction in the middle again the research question is missing.
- Line-60-62, what authors think because PVA is been used in drug delivery quite frequently, is this rationale is enough? I am not convinced please explain why PVA? And why indomethacin?
- Why it is necessary to add too much details of PVA? Is really Table 1 needed here, I would suggest to add this to Material section, when authors have a designated section then why its appearing at other places, please explain if author think otherwise.
- The rationale is completely missing?
- Proofreading is strongly advised.
Results and discussion:
- Results are descriptive please can authors focused on discussion keeping in view the manuscript rationale backed up by relevant theories and literature.
- Proofreading is strongly advised.
Conclusions:
- The major proportion of results have been used again, it really missed SO WHAT question, please focus on how your results are useful for the scientific community.
- Proofreading is strongly advised.
References:
- This section requires consistency, you are giving DOI’s of some references, either provide for all or delete the some.
Round 2
Reviewer 2 Report
My major concerns are now been addressed by the authors. However, I will still advise and English proof-reading.